# Cohort study of cardiovascular safety of different COVID-19 vaccination doses among 46 million adults in England

Samantha Ip [1,2,3,21] ✉, Teri-Louise North [4,21], Fatemeh Torabi [5], Yangfan Li[1,2,3], Hoda Abbasizanjani [5], Ashley Akbari [5], Elsie Horne[4], Rachel Denholm [4,6,7], Spencer Keene[1,3], Spiros Denaxas[8,9,10,11,12], Amitava Banerjee [9], Kamlesh Khunti[13], Cathie Sudlow [12], William N. Whiteley [12,14], Jonathan A. C. Sterne [4,6,7,22], Angela M. Wood [1,3,12,15,16,17,18,22], Venexia Walker [4,19,20,22], the CVD-COVID-UK/COVID-IMPACT Consortium* & the Longitudinal Health and Wellbeing COVID-19 National Core Study*

The first dose of COVID-19 vaccines led to an overall reduction in cardiovascular events, and in rare cases, cardiovascular complications. There is less information about the effect of second and booster doses on cardiovascular diseases. Using longitudinal health records from 45.7 million adults in England between December 2020 and January 2022, our study compared the incidence of thrombotic and cardiovascular complications up to 26 weeks after first, second and booster doses of brands and combinations of COVID-19 vaccines used during the UK vaccination program with the incidence before or without the corresponding vaccination. The incidence of common arterial thrombotic events (mainly acute myocardial infarction and ischaemic stroke) was generally lower after each vaccine dose, brand and combination. Similarly, the incidence of common venous thrombotic events, (mainly pulmonary embolism and lower limb deep venous thrombosis) was lower after vaccination. There was a higher incidence of previously reported rare harms after vaccination: vaccine-induced thrombotic thrombocytopenia after first ChAdOx1 vaccination, and myocarditis and pericarditis after first, second and transiently after booster mRNA vaccination (BNT-162b2 and mRNA-1273). These findings support the wide uptake of future COVID-19 vaccination programs.

SARS-CoV-2 vaccination prevented 14.4 million deaths from COVID-19 worldwide in the first year of the pandemic[1]. In England, which entered its third COVID-19 vaccine season in autumn 2023[2], around 90% of people aged ≥12 years have been vaccinated at least once[3]. COVID-19 vaccines are associated with rare cardiovascular complications: mRNA-based brands (e.g. BNT-162b2, mRNA1273) with myocarditis and adenovirus-based brands (e.g. ChAdOx1) with vaccine-induced thrombotic thrombocytopenia (VITT), leading to intracranial venous thrombosis (ICVT) and thrombocytopenia[4,5]. It is important to

understand the risk of thrombotic and cardiovascular complications arising from second and subsequent doses, within the general population and subpopulations[6].

In this work, we use whole population longitudinal electronic health records from 45.7 million adults in England to quantify associations of first, second, and booster doses of mRNA and non-mRNA COVID-19 vaccine brands with subsequent thrombotic and cardiovascular events. Data were accessed within the NHS England Secure Data Environment (NHSE SDE)[7], encompassing primary care, hospital

---

A full list of affiliations appears at the end of the paper. *Lists of authors and their affiliations appear at the end of the paper. ✉e-mail: hyi20@cam.ac.uk

admissions, COVID-19 testing and vaccination data, dispensed medication records in primary care data, and the Office of National Statistics death registrations. We use Cox regression to estimate adjusted hazard ratios (aHRs) and corresponding 95% confidence intervals (95% CIs) in time intervals since vaccination, adjusted for a wide range of co-morbidities, age, sex, and prior COVID-19. We show that the incidence of thrombotic and cardiovascular complications was generally lower after each dose of each vaccine brand, except for previously recognised rare complications of the ChAdOx1 vaccine and the mRNA vaccines.

## Results

### Characteristics of the study populations for first, second and booster vaccination

Over the study period from 8th December 2020 to 23rd January 2022, 45.7 million individuals met the eligibility criteria for our first-dose analyses (Table 1, Supplementary Fig. 1). Among these, 37.3 million people received a first ChAdOx1, BNT-162b2 or mRNA1273 vaccination and were eligible for the second dose analyses (Supplementary Table 1). We refer to the first and second dose vaccinations as the "primary course", distinguishing them from booster vaccine doses. Third dose vaccination, which is distinct from the booster dose and is administered as part of an extended primary course, was not considered. Of those who received a first dose, 35.9 million people received ChAdOx1, BNT-162b2 or mRNA1273 primary course vaccinations and were eligible for analyses of booster vaccination (Supplementary Table 2).

Differences in characteristics of people in the three vaccination cohorts reflect known vaccination uptake profiles. Considering the strong similarity in characteristics of the second and booster dose cohorts, we compared characteristics of the second and booster dose cohorts with the first dose cohort. Compared to the first dose cohort (Table 1), people in the second and booster vaccine dose cohorts (Supplementary Tables 1 and 2), who had received at least one COVID-19 vaccine dose, were older (<40 years: 31% vs 38%), and less likely to have a non-White ethnicity (16% vs 20%), or be deprived (IMD deciles 1–4, 36% vs 39%). They were modestly more likely to have cancer (18–19% vs 14%), to have contracted COVID-19 previously (7–8% vs 6%), to take blood pressure (22% vs 18%) or lipid-lowering medication (17–18% vs 15%), or to be clinically vulnerable (25% vs 22%).

Characteristics of people receiving different vaccine brands reflect vaccine availability during the UK vaccine rollout, which was in order of priority groups specified by the Joint Committee on Vaccination and Immunisation (JCVI). People younger than 40 years old and not in a priority group were offered only mRNA (BNT-162b2 or mRNA1273) vaccine brands[8].

### Incidence of different cardiovascular events

From the start of vaccine rollout (8th December 2020) up to first vaccination, during approximately 21 million person-years there were 75,655 arterial and 21,230 venous incident thrombotic events (Table 2 and Supplementary Table 4). Arterial thromboses included acute myocardial infarction (AMI; 37,915) and ischaemic stroke (36,720). Venous thromboses included pulmonary embolism (PE; 11,835) and lower limb deep venous thrombosis (DVT; 9075), intracranial venous thrombosis (ICVT; 370) and portal vein thrombosis (PVT; 185). Other incident cardiovascular events included subarachnoid haemorrhage and haemorrhagic stroke (SAH; 5235), mesenteric thrombus (1515), thrombocytopenia (1885), myocarditis (590) and pericarditis (455). The numbers of events and incidence rates after first, second and booster vaccinations reflected the older age of vaccinated, compared with unvaccinated, people (Supplementary Tables 4–7).

### COVID-19 vaccination and arterial, venous, and other thrombotic events

We used Cox models to estimate adjusted hazard ratios (aHRs) and corresponding 95% CIs, comparing the incidence of thrombotic and cardiovascular events after first, second and booster vaccine doses with the incidence before or without the corresponding vaccine dose, adjusting for a wide range of potential confounding factors (Supplementary Tables 8–25). Associations with second dose were estimated in people who had a first dose, and associations with booster vaccination in people who had both first and second vaccinations. The aHRs after first doses of mRNA-1273 were imprecisely estimated, because this brand was received by fewer people who were generally younger, so were reported in supplementary tables but omitted from the figures.

The incidence of composite arterial thrombotic events (AMI, ischaemic stroke and other arterial embolism) was similar or lower after first, second and booster doses of ChAdOx1 and BNT-162b2 vaccines and booster doses of mRNA-1273, compared to follow-up before or without the corresponding vaccine dose in eligible people (Fig. 1, Supplementary Tables 8–25). After second and booster vaccine doses, the reduction in incidence of composite arterial thrombosis was greater than after first vaccination. For example, aHRs for arterial thrombotic events 13–24 weeks after first vaccine dose were 0.99 (95% CI 0.97–1.02) after ChAdOx1 and 0.90 (0.88–0.93) after BNT-162b2. Corresponding aHRs after second doses were 0.73 (0.70–0.76) and 0.80 (0.77–0.83) respectively. aHRs for mRNA booster vaccination after primary course of ChAdOx1 were 0.71 (0.66–0.76) 13–24 weeks after BNT-162b2 and 0.67 (0.62–0.72) 5–24 weeks after mRNA-1273 respectively. After a primary course of BNT-162b2, aHRs 13–24 weeks after booster vaccination were 0.73 (0.69–0.77) and 1.21 (0.38–3.86) after BNT-162b2 and mRNA-1273 respectively. For all vaccine brands and doses, aHRs in the first few weeks were lower than in later weeks. The aHR profiles for AMI and ischaemic stroke were similar to those of composite arterial thrombosis, for all vaccine brands and doses.

Similar to arterial events, the incidence of composite venous thrombotic events (PE, DVT, ICVT and PVT) was generally lower after first, second and booster doses vaccination, compared to follow-up before or without the corresponding vaccine dose (Fig. 2, Supplementary Tables 8–25). After second and booster vaccine doses, the reduction in incidence of composite venous thrombotic events was greater than after first vaccination. For example, aHRs 13–24 weeks after first vaccine doses were 0.94 (95%CI 0.90–0.98) after ChAdOx1 and 0.85 (0.81–0.88) after BNT-162b2. Corresponding aHRs after second doses were 0.68 (0.63–0.73) and 0.77 (0.72–0.83) respectively. aHRs for mRNA booster vaccination after primary course of ChAdOx1 were 0.63 (0.54–0.74) 13–24 weeks after BNT-162b2 and 0.55 (0.47–0.65) 5–24 weeks after mRNA-1273. After a primary course of BNT-162b2, aHRs for booster vaccination were 0.56 (0.49–0.63) 13–24 weeks after BNT-162b2 and 0.58 (0.45–0.74) 5–24 weeks after mRNA-1273. The aHR profiles for common venous events PE and DVT were similar to those of composite venous thrombosis for all vaccine brands and doses. In contrast, there was a higher incidence of the known rare complication ICVT immediately after first dose of ChAdOx1 with greatest aHR 2 weeks after vaccination (5.92; 95% CI 4.07–8.63). There was no increase in incidence of ICVT after second dose of ChAdOx1 (Supplementary Table 8), or after any other vaccine brand.

There was a higher incidence of thrombocytopenia after first dose of ChAdOx1 (Fig. 3, Supplementary Table 8), compared with no vaccination, with greatest aHR 2 weeks after vaccination (2.07; 95% CI 1.67–2.58) but no increase in incidence after second dose of ChAdOx1 or after first or second dose of BNT-162b2, or after booster dose of mRNA-1273 following primary course of ChAdOx1 (Supplementary Tables 9, 11 and 12). There was a higher incidence of thrombocytopenia 13–24 weeks after a booster dose of BNT-162b2 following primary course of ChAdOx1 (aHR 2.16; 95% CI 1.26–3.69) (Fig. 3, Supplementary Table 15).

The incidence of SAH and mesenteric thrombus was similar or lower after all doses and brands of vaccination (Supplementary Tables 8–25), compared with before or without the corresponding dose. The incidence of mesenteric thrombus and SAH were markedly lower beyond 4 weeks after second dose of ChAdOx1 or BNT-162b2

**Table 1 | Characteristics (number [%]) of the population under study – dose 1 cohort**

| Characteristic | | All eligible for dose 1 analysis | Vaccination status during dose 1 follow-up | | | |
| --- | --- | --- | --- | --- | --- | --- |
| | | | Unvaccinated | ChAdOx1 | BNT162b2 | mRNA-1273 |
| Total | All | 45,673,965 | 8,424,120 (18.44%) | 19,317,985 (42.30%) | 16,846,995 (36.89%) | 1,084,865 (2.38%) |
| Sex | Male | 22,607,555 (49.5%) | 4,718,635 (56.01%) | 9,421,000 (48.77%) | 7,875,835 (46.75%) | 592,085 (54.58%) |
| | Female | 23,066,410 (50.5%) | 3,705,480 (43.99%) | 9,896,985 (51.23%) | 8,971,165 (53.25%) | 492,780 (45.42%) |
| Age (years) | 18–29 | 8,847,920 (19.4%) | 2,776,200 (32.96%) | 1,021,805 (5.29%) | 4,572,045 (27.14%) | 477,870 (44.05%) |
| | 30–39 | 8,484,105 (18.6%) | 2,450,210 (29.09%) | 1,630,695 (8.44%) | 4,020,215 (23.86%) | 382,985 (35.30%) |
| | 40–49 | 7,540,600 (16.5%) | 1,430,495 (16.98%) | 4,322,935 (22.38%) | 1,585,815 (9.41%) | 201,355 (18.56%) |
| | 50–59 | 7,700,005 (16.9%) | 865,605 (10.28%) | 5,284,090 (27.35%) | 1,534,230 (9.11%) | 16,080 (1.48%) |
| | 60–69 | 5,871,400 (12.9%) | 474,340 (5.63%) | 3,762,510 (19.48%) | 1,629,755 (9.67%) | 4795 (0.44%) |
| | 70–79 | 4,594,010 (10.1%) | 248,830 (2.95%) | 2,529,425 (13.09%) | 1,814,400 (10.77%) | 1355 (0.12%) |
| | 80–89 | 2,157,220 (4.72%) | 130,310 (1.55%) | 575,940 (2.98%) | 1,450,595 (8.61%) | 375 (0.03%) |
| | 90+ | 478,710 (1.05%) | 48,130 (0.57%) | 190,590 (0.99%) | 239,940 (1.42%) | 50 (0.00%) |
| Ethnic group | Asian | 3,740,455 (8.19%) | 855,560 (10.16%) | 1,316,815 (6.82%) | 1,488,545 (8.84%) | 79,535 (7.33%) |
| | Black | 1,675,070 (3.67%) | 651,935 (7.74%) | 489,720 (2.54%) | 497,755 (2.95%) | 35,660 (3.29%) |
| | Mixed | 717,695 (1.57%) | 249,425 (2.96%) | 196,505 (1.02%) | 248,795 (1.48%) | 22,970 (2.12%) |
| | Other | 1,448,035 (3.17%) | 626,600 (7.44%) | 361,545 (1.87%) | 422,270 (2.51%) | 37,620 (3.47%) |
| | White | 36479155 (79.9%) | 5285020 (62.74%) | 16,524,935 (85.54%) | 13,798,605 (81.91%) | 870,595 (80.25%) |
| | Unknown/Missing | 1,613,555 (3.53%) | 755,580 (8.97%) | 428,465 (2.22%) | 391,025 (2.32%) | 38,485 (3.55%) |
| Index of multiple deprivation (1 most, 10 leas deprived) | 1–2 | 8,651,540 (18.9%) | 2,294,285 (27.23%) | 3,253,630 (16.84%) | 2,926,715 (17.37%) | 176,910 (16.31%) |
| | 3–4 | 9,015,905 (19.7%) | 2,090,015 (24.81%) | 3,535,865 (18.30%) | 3,184,335 (18.90%) | 205,690 (18.96%) |
| | 5–6 | 8,815,855 (19.3%) | 1,554,155 (18.45%) | 3,803,565 (19.69%) | 3,258,740 (19.34%) | 199,395 (18.38%) |
| | 7–8 | 8,580,765 (18.8%) | 1,198,325 (14.22%) | 3,948,195 (20.44%) | 3,240,875 (19.24%) | 193,370 (17.82%) |
| | 9–10 | 8,300,280 (18.2%) | 922,005 (10.94%) | 4,037,545 (20.90%) | 3,143,075 (18.66%) | 197,655 (18.22%) |
| | Missing | 2,309,620 (5.06%) | 365,330 (4.34%) | 739,185 (3.83%) | 1,093,260 (6.49%) | 111,845 (10.31%) |
| Smoking status | Current | 7,788,670 (17.1%) | 2,216,230 (26.31%) | 3,000,925 (15.53%) | 2,385,020 (14.16%) | 186,495 (17.19%) |
| | Former | 10,414,755 (22.8%) | 1,267,720 (15.05%) | 5,255,825 (27.21%) | 3,728,875 (22.13%) | 162,335 (14.96%) |
| | Never | 25,332,265 (55.5%) | 4,158,860 (49.37%) | 10,754,915 (55.67%) | 9,790,240 (58.11%) | 628,250 (57.91%) |
| | Missing | 2,138,275 (4.68%) | 781,300 (9.27%) | 306,325 (1.59%) | 942,865 (5.60%) | 107,785 (9.94%) |
| Medical history | AMI | 1,161,760 (2.54%) | 85,295 (1.01%) | 567,515 (2.94%) | 507,980 (3.02%) | 970 (0.09%) |
| | Diabetes | 3,907,740 (8.56%) | 301,965 (3.58%) | 1,999,260 (10.35%) | 1,598,265 (9.49%) | 8250 (0.76%) |
| | Depression | 9,426,550 (20.6%) | 1,325,580 (15.74%) | 4,602,700 (23.83%) | 3,328,455 (19.76%) | 169,815 (15.65%) |
| | Obesity | 4,384,350 (9.60%) | 402,695 (4.78%) | 2,298,955 (11.90%) | 1,639,185 (9.73%) | 43,515 (4.01%) |
| | Cancer | 6,465,720 (14.2%) | 576,160 (6.84%) | 3,122,945 (16.17%) | 2,664,130 (15.81%) | 102,485 (9.45%) |
| | COPD | 1,549,765 (3.39%) | 116,345 (1.38%) | 798,160 (4.13%) | 633,205 (3.76%) | 2055 (0.19%) |
| | Liver disease | 225,630 (0.49%) | 32,055 (0.38%) | 117,550 (0.61%) | 75,170 (0.45%) | 855 (0.08%) |
| | CKD | 2,867,785 (6.28%) | 214,160 (2.54%) | 1,346,545 (6.97%) | 1301275 (7.72%) | 5805 (0.54%) |
| | Dementia | 510,600 (1.12%) | 49,230 (0.58%) | 268,085 (1.39%) | 193,075 (1.15%) | 210 (0.02%) |
| | All stroke | 835,305 (1.83%) | 67,115 (0.80%) | 423,070 (2.19%) | 344,140 (2.04%) | 980 (0.09%) |
| | All VTE | 642,035 (1.41%) | 66,330 (0.79%) | 329,515 (1.71%) | 244,180 (1.45%) | 2010 (0.19%) |
| | Thrombophilia | 43,440 (0.10%) | 5130 (0.06%) | 20,795 (0.11%) | 17,020 (0.10%) | 495 (0.05%) |
| Major surgery in the last year | | 3,438,335 (9.23%) | 423,490 (5.03%) | 732,470 (5.17%) | 1,602,550 (9.51%) | 53,680 (4.95%) |
| No. of unique diseases in the last year | 0 | 36,130,080 (79.1%) | 7,478,345 (88.77%) | 14,694,520 (76.07%) | 13,005,675 (77.20%) | 951,540 (87.71%) |
| | 1–5 | 9,412,710 (20.6%) | 933,460 (11.08%) | 4,554,595 (23.58%) | 3,791,520 (22.51%) | 133,135 (12.27%) |
| | 6+ | 131,170 (0.29%) | 12,300 (0.15%) | 68,875 (0.36%) | 49,805 (0.30%) | 190 (0.02%) |
| Region | Northwest | 5,658,370 (12.4%) | 1,006,965 (11.95%) | 2,423,155 (12.54%) | 2,106,490 (12.50%) | 121,760 (11.22%) |
| | Southeast | 6,574,915 (14.4%) | 1,007,385 (11.96%) | 3,002,770 (15.54%) | 2,378,170 (14.12%) | 186,590 (17.20%) |
| | London | 7,210,945 (15.8%) | 2,248,485 (26.69%) | 2,356,510 (12.20%) | 2,429,115 (14.42%) | 176,835 (16.30%) |
| | East of England | 4,040,560 (8.85%) | 663,225 (7.87%) | 1,830,055 (9.47%) | 1,455,715 (8.64%) | 91,565 (8.44%) |
| | Southwest | 3,574,255 (7.83%) | 463,450 (5.50%) | 1,616,385 (8.37%) | 1,397,850 (8.30%) | 96,570 (8.90%) |
| | Yorkshire/Humber | 5,096,465 (11.2%) | 714,715 (8.48%) | 1,900,555 (9.84%) | 1,517,425 (9.01%) | 113,245 (10.44%) |
| | East Midlands | 4,245,940 (9.30%) | 484,935 (5.76%) | 1,434,740 (7.43%) | 1,192,780 (7.08%) | 13,055 (1.20%) |
| | West Midlands | 3,125,510 (6.84%) | 821,550 (9.75%) | 1,942,855 (10.06%) | 1,627,415 (9.66%) | 70,815 (6.53%) |
| | Northeast | 4,462,635 (9.77%) | 254,925 (3.03%) | 761,420 (3.94%) | 628,785 (3.73%) | 39,240 (3.62%) |
| | Missing | 1,684,370 (3.69%) | 758,495 (9.00%) | 2,049,535 (10.61%) | 2,113,255 (12.54%) | 175,180 (16.15%) |

**Table 1 (continued) | Characteristics (number [%]) of the population under study – dose 1 cohort**

| Characteristic | | All eligible for dose 1 analysis | | Vaccination status during dose 1 follow-up | | |
|---|---|---|---|---|---|---|
| | | | Unvaccinated | ChAdOx1 | BNT162b2 | mRNA-1273 |
| Prior COVID-19 at index date | | 2,496,910 (5.5%) | 119,535 (1.42%) | 464,180 (2.40%)) | 488,985 (2.90%)) | 29,215 (2.69%) |
| Medications taken in the last 3 months | Antiplatelet | 119,325 (1.42%) | 142,340 (1.00%) | 1,215,145 (6.29%) | 1,108,890 (6.58%) | 2445 (0.23%) |
| | Blood pressure lowering | 8,390,090 (18.4%) | 367,775 (4.37%) | 4,520,250 (23.40%) | 3,486,990 (20.70%) | 15,075 (1.39%) |
| | Lipid lowering | 6,651,900 (14.6%) | 241,945 (2.87%) | 3,552,420 (18.39%) | 2,851,365 (16.93%) | 6170 (0.57%) |
| | Anticoagulant | 1,300,380 (2.85%) | 64,025 (0.76%) | 613,405 (3.18%) | 622,200 (3.69%) | 750 (0.07%) |
| | COCP | 612,760 (1.34%) | 67,420 (0.80%) | 143,415 (0.74%) | 368,470 (2.19%) | 33,455 (3.08%) |
| | HRT | 530,515 (1.16%) | 20,320 (0.24%) | 350,180 (1.81%) | 156,765 (0.93%) | 3250 (0.30%) |
| Clinically vulnerable | Neither | 35,461,085 (77.6%) | 7,547,780 (89.60%) | 14,122,735 (73.11%) | 12,731,880 (75.57%)) | 1,058,690 (97.59%) |
| | Vulnerable | 8,084,085 (17.7%) | 711,505 (8.45%) | 4,094,310 (21.19%) | 3,255,215 (19.32%) | 23,055 (2.13%) |
| | Extremely vulnerable | 2,128,795 (4.66%) | 164,830 (1.96%) | 1,100,945 (5.70%) | 859,900 (5.10%) | 3120 (0.29%) |

*CKD* Chronic Kidney Disease, *COCP* Combined Oral Contraceptive Pill, *COPD* Chronic Obstructive Pulmonary Disease, *HRT* Hormone Replacement Therapy

vaccines. For example, aHRs for mesenteric thrombus 13–24 weeks after second dose were 0.64 (95% CI 0.49–0.83) for ChAdOx1 and 0.64 (0.49–0.83) for BNT-162b2. Corresponding aHRs for SAH 13–24 weeks after second dose were 0.64 (0.56–0.74) for ChAdOx1 and 0.84 (0.73–0.98) for BNT-162b2.

The incidence of myocarditis was higher after first dose of BNT-162b2 vaccine, with greatest aHR 1 week after vaccination (2.05; 95% CI 1.28–3.29) (Fig. 3; Supplementary Table 9), higher one week after second dose of BNT-162b2 (aHR 3.14; 95% CI 2.04–4.85) (Supplementary Table 12) and higher after some mRNA booster vaccinations (for example, the aHR 1 week after booster BNT-162b2 vaccination was 1.65; 95% CI 1.07–2.57 following primary course of any of ChAdOx1, BNT-162b2 or mRNA-1273) (Supplementary Tables 19, 21, 22, 24, and 25). Otherwise, the incidence of myocarditis after vaccination was similar to or lower than that before or without vaccination.

The incidence of pericarditis was higher after first dose of ChAdOx1, with greatest aHR 2 weeks after vaccination (1.74; 95% CI 1.04–2.91), after first dose of BNT-162b2 (for example, the aHR 13–24 weeks after vaccination was 1.50; 95% CI 1.17–1.92) (Supplementary Tables 8 and 9), after second dose of BNT-162b2 (aHR 3–4 weeks after vaccination 2.42; 95% CI 1.62–3.62) (Supplementary Table 12), and after mRNA-based booster vaccination (for example, the aHR 2 weeks after booster dose of BNT-162b2 vaccine following primary course of any vaccine brand was 1.73; 95% CI 1.05–2.83) (Supplementary Tables 22 and 25). Otherwise, the incidence of pericarditis after vaccination was similar to, or lower than, that before or without vaccination, with a halving in incidence beyond 4 weeks after second dose of ChAdOx1 (for example, the aHR after 13–24 weeks was 0.32; 95% CI 0.19–0.51) (Supplementary Table 11).

When follow-up was censored at the time of the US Centers for Disease Control (CDC) public announcement on myocarditis and pericarditis on 17th May 2021, the incidence of myocarditis and pericarditis after both first and second doses of ChAdOx1 and BNT-162b2 was similar to that before or without such vaccinations. One exception was a higher incidence of pericarditis 5–24 weeks after first dose BNT-162b2 vaccination (aHR 1.89; 95% CI 1.13–3.16) (Supplementary Fig. 8; Supplementary Table 30). Note that the smaller number of events in this sensitivity analyses meant that aHRs were less precisely estimated than in the main analyses.

**COVID-19 vaccination and thrombotic events within population subgroups**

Subgroup analyses by age group, ethnic group, previous history of COVID-19, history of the event of interest and sex were conducted for composite arterial and composite venous outcomes (Supplementary Figs. 2–7; Supplementary Tables 26–29). Associations between vaccination and thrombotic events were generally similar across subgroups, with the following exceptions. For composite arterial events after first doses of ChAdOx1 or BNT-162b2, aHRs in people with unknown ethnicity were higher than in people with known ethnicity (Supplementary Figs. 2 and 4; Supplementary Tables 26 and 27). In general, aHRs for composite arterial and composite venous events after first and second doses of ChAdOx1 or BNT-162b2 were higher in males than females (Supplementary Figs. 2–5).

## Discussion

This study used whole population longitudinal health records from over 45.7 million adults in England to quantify associations of first, second and booster doses of COVID-19 vaccine brands used during the first two years of the UK vaccine rollout with the incidence of arterial and venous thromboses, thrombocytopenia and myocarditis. Estimated hazard ratios were adjusted for a wide range of potential confounders. The incidence of thrombotic and cardiovascular complications was generally lower after each dose of each vaccine brand. Exceptions, consistent with previous findings that have been recognised by medicines regulators, included rare complications of the ChAdOx1 vaccine (ICVT and thrombocytopenia, due to vaccine-induced immune thrombocytopenia and thrombosis) and the mRNA vaccines (myocarditis and pericarditis). There were few differences between subgroups defined by demographic and clinical characteristics. These findings, in conjunction with the long-term higher risk of severe cardiovascular and other complications associated with COVID-19, offer compelling evidence supporting the net cardiovascular benefit of COVID vaccination.

The strengths of this study lie in the representativeness of the whole population data, offering an overview of thrombotic events after vaccination, as well as the comprehensive analyses of different vaccine dose and brand combinations in the general population. Consequently, the findings should apply to nations with comparable demographics and healthcare systems. The very large sample size facilitated estimation of associations with rare outcomes, within time periods after vaccination, and within population subgroups. The extensive longitudinal coverage of the health records also allowed examination of events after the first, second, and booster vaccinations. We addressed potential confounding by adjusting for a wide-range of demographic factors and prior diagnoses available in primary and secondary care records, defined using clinician-validated code lists

**Table 2 | Number of events, person-years and incidence rates for composite arterial and venous thrombotic events for first, second and booster vaccinations[a]**

| Vaccination dose | Brand | Event | Events/100K person-years | | Incidence rate | |
|---|---|---|---|---|---|---|
| **Dose 1** | | | **No vaccination** | **After first dose** | **No vaccination** | **After first dose** |
| | **ChAdOx1** | Arterial | 75,655/205.00 | 73,330/ 95.63 | 369.05 | 766.83 |
| | | Venous | 21,230/205.00 | 17,525/ 96.09 | 103.56 | 182.39 |
| | **BNT-162b2** | Arterial | 75,655/205.00 | 57,050/ 80.99 | 369.05 | 704.38 |
| | | Venous | 21,230/205.00 | 11,860/ 81.00 | 103.56 | 146.42 |
| | **mRNA-1273** | Arterial | 75,655/205.00 | 255/ 5.11 | 369.05 | 49.89 |
| | | Venous | 21,230/205.00 | 190/ 5.10 | 103.56 | 37.25 |
| **Doses 1 and 2** | | | **After first dose** | **After second dose** | **After first dose** | **After second dose** |
| | **ChAdOx1** | Arterial | 39,425/41.23 | 70,485/93.36 | 956.17 | 755.01 |
| | | Venous | 8685/41.27 | 16,120/93.60 | 210.46 | 172.22 |
| | **BNT-162b2** | Arterial | 26,975/33.63 | 55,620/72.44 | 802.22 | 767.8 |
| | | Venous | 5260/33.67 | 11,400/72.69 | 156.22 | 156.83 |
| | **mRNA-1273** | Arterial | 160/2.17 | 200/4.06 | 73.66 | 49.24 |
| | | Venous | 90/2.18 | 135/4.06 | 41.37 | 33.29 |
| **Booster** | | | **Primary course only** | **After booster vaccination** | **Primary course only** | **After booster vaccination** |
| **Primary course of ChAdOx1 or BNT-162b2 or mRNA-1273** | **BNT-162b2** | Arterial | 157,235/182.21 | 39,735/38.72 | 862.93 | 1026.14 |
| | | Venous | 32,545/182.54 | 7855/ 38.97 | 178.29 | 201.54 |
| | **mRNA-1273** | Arterial | 157,235/182.22 | 3200/ 7.46 | 862.88 | 429.07 |
| | | Venous | 32,545/182.56 | 760/ 7.47 | 178.27 | 101.79 |
| | **BNT-162b2 or mRNA-1273** | Arterial | 157,235/182.21 | 42,935/ 46.20 | 862.93 | 929.36 |
| | | Venous | 32,545/182.58 | 8615/ 46.37 | 178.25 | 185.79 |
| **Primary course of BNT-162b2 or mRNA-1273** | **BNT-162b2** | Arterial | 70,215/80.31 | 21,425/19.04 | 874.28 | 1125.34 |
| | | Venous | 13,830/80.45 | 3975/19.15 | 171.91 | 207.62 |
| | **mRNA-1273** | Arterial | 70,215/80.29 | 870/ 2.23 | 874.52 | 389.78 |
| | | Venous | 13,830/80.45 | 185/ 2.24 | 171.91 | 82.76 |
| | **BNT-162b2 or mRNA-1273** | Arterial | 22,300/21.25 | 874.73 | 1049.33 | 22,300/21.25 |
| | | Venous | 4165/21.39 | 171.88 | 194.72 | 4165/21.39 |
| **Primary course of ChAdOx1** | **BNT-162b2** | Arterial | 87,020/101.93 | 18,310/ 19.76 | 853.73 | 926.59 |
| | | Venous | 18,715/102.13 | 3880/ 19.81 | 183.25 | 195.83 |
| | **mRNA-1273** | Arterial | 87,020/101.95 | 2325/ 5.21 | 853.59 | 446.61 |
| | | Venous | 18,715/102.12 | 575/ 5.24 | 183.27 | 109.82 |
| | **BNT-162b2 or mRNA-1273** | Arterial | 87,020/101.95 | 20,635/ 24.94 | 853.54 | 827.32 |
| | | Venous | 18,715/102.08 | 4455/ 25.08 | 183.33 | 177.61 |
| **Primary course of BNT-162b2** | **BNT-162b2** | Arterial | 70,005/76.60 | 21,415/18.78 | 913.88 | 1140.44 |
| | | Venous | 13,700/76.75 | 3965/18.91 | 178.51 | 209.65 |
| | **mRNA-1273** | Arterial | 70,005/76.59 | 860/ 2.02 | 914 | 426.6 |
| | | Venous | 13,700/76.80 | 180/ 2.04 | 178.38 | 88.15 |
| | **BNT-162b2 or mRNA-1273** | Arterial | 70,005/76.59 | 22,275/20.79 | 913.99 | 1071.28 |
| | | Venous | 13,700/76.73 | 4145/20.96 | 178.54 | 197.8 |

[a]All counts rounded to the nearest 5 and counts less than 10 displayed as '10'. Bold formatting identifies headers within the table, specifically at each vaccination dose, to clearly delineate comparison groups.

that are accessible via our GitHub repository (available at https://github.com/BHFDSC/CCU002_06). Our analysis adhered to a pre-specified protocol with one deviation: we censored all analyses 26 weeks after vaccination to avoid interference from subsequent vaccinations[9].

This study has several limitations. First, residual confounding, including that linked to delayed vaccination in high-risk individuals, may persist despite extensive adjustments for available covariates. We were able to identify some, but not all people who were clinically vulnerable (and hence might have been eligible for earlier vaccination): for example, younger adults in long-stay settings could not be reliably identified. Second, we did not adjust for potential confounding by time-varying post-baseline factors that may have influenced receipt of

vaccination and the outcomes of interest: for example, development of respiratory symptoms or being admitted into hospital leading to postponement of vaccination. Such confounding may explain estimated lower hazard ratios soon after vaccination[10]. Third, ascertainment of some outcomes may have been influenced by public announcements from regulatory agencies, such as the European Medicines Agency Pharmacovigilance Risk Assessment Committee announcement[11] or the CDC announcement on myocarditis[12,13]. This was addressed in sensitivity analyses for myocarditis and pericarditis, censoring follow-up at the time of public announcements of these adverse effects of vaccination, although the shorter follow-up times and corresponding smaller numbers of events in the restricted analyses meant that aHRs were estimated with reduced precision. Fourth,

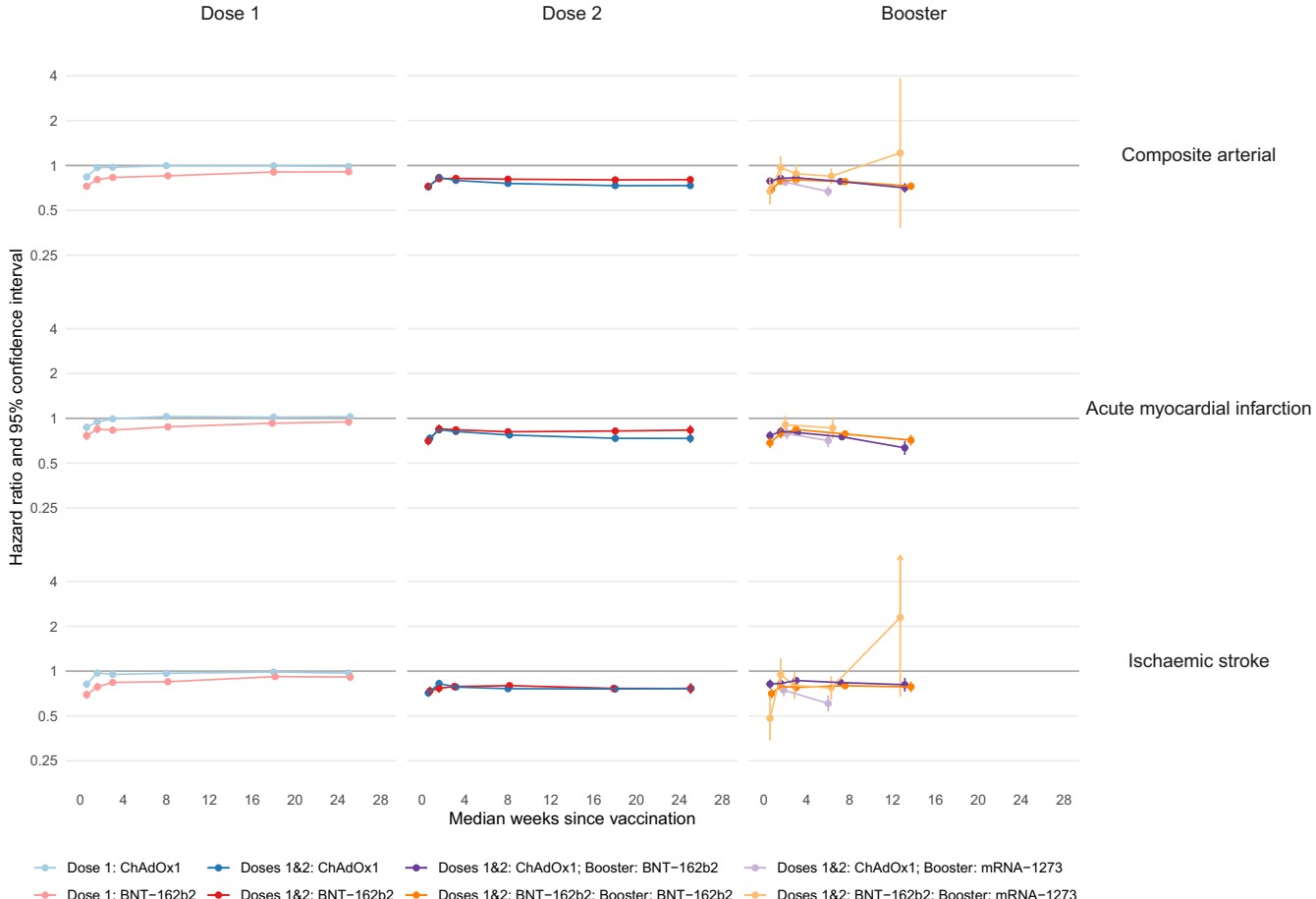

**Fig. 1 | Adjusted hazard ratios (aHRs) and 95% confidence intervals (95% CIs) for arterial thromboses following COVID-19 vaccination, by dose and brand.** Vertical lines depict 95% CIs; these are not visible when they are very narrow. There were no AMI events during weeks 13–26 after mRNA-1273 booster vaccination, so follow-up is grouped as 1–4 and 5–26 weeks post-vaccination. The number of people eligible for first, second, and booster dose analyses were 45,673,965; 37,249,850 and 35,853,120, respectively. The number of people who received a first dose of ChAdOx1, BNT-162b2 and mRNA-1273 were 19,317,985, 16,846,995, and 1,084,865, respectively; a second dose of ChAdOx1, BNT-162b2 and mRNA-1273 were 18,920,225, 15,961,330, and 971,565, respectively; a booster dose of BNT-162b2 and mRNA-1273 following a primary course of ChAdOx1 were 11,964,635 and 4,153,760 respectively; a booster dose of BNT-162b2 and mRNA-1273 following a primary course of BNT-162b2 were 9,821,835 and 1,914,925, respectively. The numerical values of hazard ratios and 95% CIs are displayed in Supplementary Tables 8, 9, 11, 12, 14, 15, 17 and 18.

outcomes may be underreported, particularly from people in nursing homes or among those with severe health conditions, due to diagnostic challenges; also, routine electronic health records, not intended for research, may under-ascertain less severe, non-hospitalised events. Both forms of potential underreporting, however, are expected to be uncommon for hospitalised thrombotic events[14]. Fifth, we restricted follow-up to 26 weeks after vaccination to prevent an influence of subsequent vaccinations on estimated associations and limit the impact of delayed vaccination on our findings. Horne et al. demonstrated selection bias in estimated HRs for non-COVID-19 death arising from deferred next-dose vaccination in people with a recent confirmed COVID-19 diagnosis or in poor health[9]. Sixth, we did not address long-term safety of vaccination, or the impact of subsequent booster doses.

The incidence of arterial and venous thrombotic events was generally lower after COVID-19 vaccination than before or without vaccination. The higher incidence of cardiovascular events after COVID-19 is well-established[14–17] and a plausible explanation tfor reductions in these events after vaccination is that vaccination prevents COVID-19, particularly severe COVID-19[18–26]. However, quantifying the mediating role of COVID-19 in cardioprotective effects of vaccination is beyond the scope of this paper.

The cohort design of our study allows estimation of incidence rates (see Table 2) and hence facilitates an understanding of the population-level impact of vaccination on common and rare cardiovascular events. However, our findings may be biased by unmeasured confounding, as discussed above. The self-controlled case series (SCCS) method aims to avoid the need to control for time-invariant confounders, but entails assumptions such as event-independent observation periods and medically-informed risk period definitions, and is particularly suited to short-term outcomes. Our results thus complement, and may be triangulated with, previous publications that examined cardiovascular implications of COVID-19 vaccination using the SCCS method[25,27,28].

This England-wide study offers reassurance regarding the cardiovascular safety of COVID-19 vaccines, with lower incidence of common cardiovascular events outweighing the higher incidence of their known rare cardiovascular complications. We found no novel cardiovascular complications or new associations with subsequent doses. Our findings support the wide uptake of future COVID-19 vaccination programs. We hope this evidence addresses public concerns, supporting continued trust and participation in vaccination programs and adherence to public health guidelines.

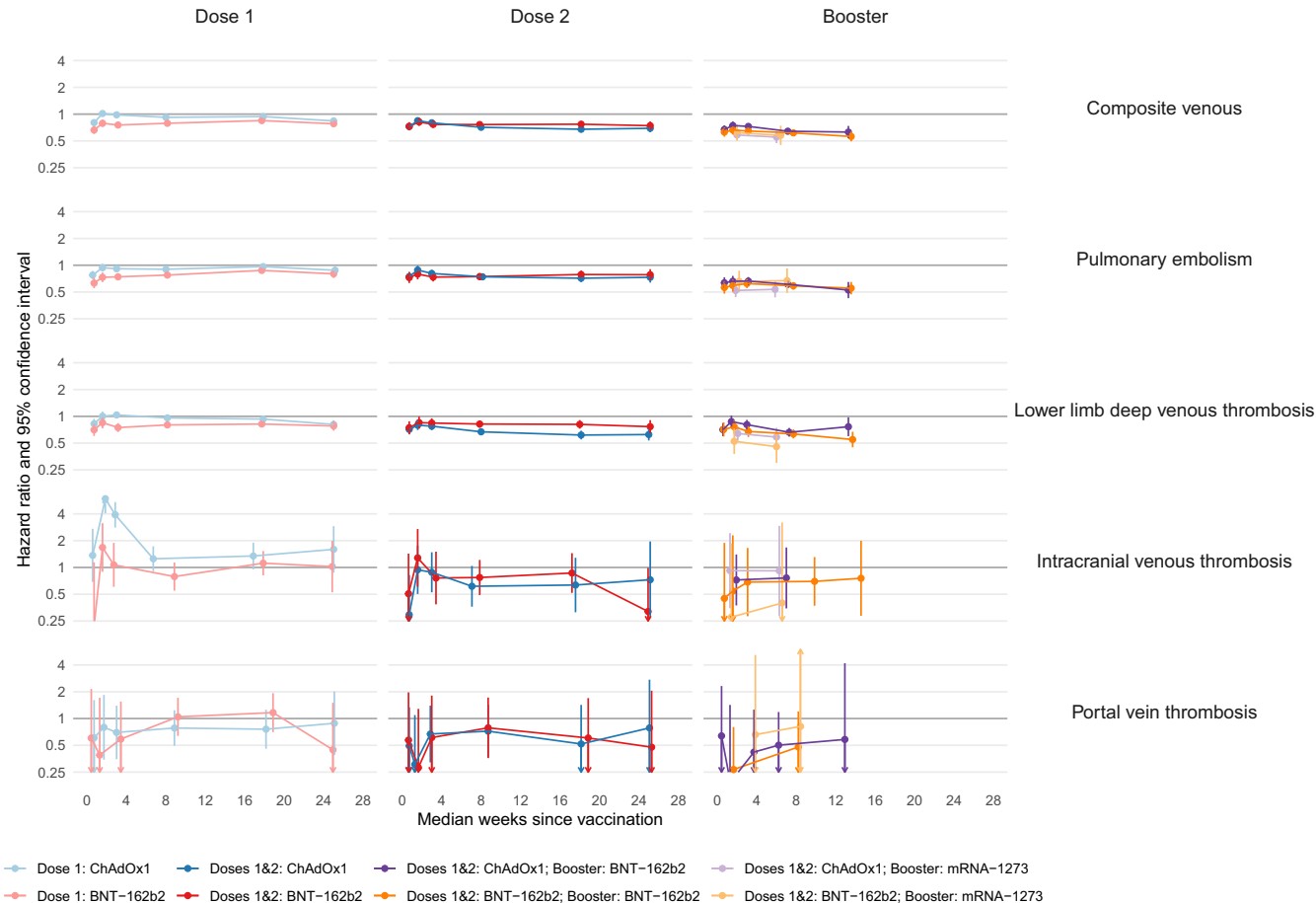

**Fig. 2 | Adjusted hazard ratios (aHRs) and 95% confidence intervals (95% CIs) for venous thromboses following COVID-19 vaccination, by dose and brand.** Vertical lines depict 95% CIs; these are not visible when they are very narrow. The number of people eligible for first, second, and booster dose analyses were 45,673,965; 37,249,850 and 35,853,120, respectively. The number of people who received a first dose of ChAdOx1, BNT-162b2 and mRNA-1273 were 19,317,985, 16,846,995, and 1,084,865, respectively; a second dose of ChAdOx1, BNT-162b2 and mRNA-1273 were 18,920,225, 15,961,330, and 971,565, respectively; a booster dose of BNT-162b2 and mRNA-1273 following a primary course of ChAdOx1 were 11,964,635 and 4,153,760 respectively; a booster dose of BNT-162b2 and mRNA-1273 following a primary course of BNT-162b2 were 9,821,835 and 1,914,925, respectively. The numerical values of hazard ratios and 95% CIs are displayed in Supplementary Tables 8, 9, 11, 12, 14, 15, 17 and 18.

## Methods

This analysis was performed according to a pre-specified analysis plan published on GitHub, along with the code lists to define variables and analysis code: https://github.com/BHFDSC/CCU002_06.

### Study population

We analysed de-identified data, made available for the BHF Data Science Centre's CVD-COVID-UK/COVID-IMPACT Consortium within the NHS England Secure Data Environment[7,29], which is a secure, privacy-protecting platform. This data consists of linked datasets including General Practice Extraction Service Extract for Pandemic Planning and Research (GDPPR), hospital admission data from Secondary Uses Service (SUS), Hospital Episode Statistics for admitted patient care (HES-APC), national laboratory COVID-19 testing data from the UK Health Security Agency (UKHSA) Second Generation Surveillance System (SGSS), Office for National Statistics (ONS) Civil Registration of Deaths (ONS deaths registry), medicines dispensed in primary care data and COVID-19 vaccination data.

The primary course of vaccination consists of the first and second vaccinations and, for certain groups such as people with severe immunosuppression, a third vaccination: this is distinct from booster vaccination that is given some time after the primary course[30].

People were included in the study if they were alive on 8th December 2020; aged 18–110 years inclusive; had a record in the GDPPR; recorded as male or female; and living in England. People with missing Lower-layer Super Output Area (LSOA) data were assumed to live in England. People were excluded if (1) they were vaccinated before 8th December 2020; (2) they were recorded as having a second dose and/or a booster or third dose, before or without records of first and second dose vaccinations respectively; (3) the interval between their first and second vaccination was less than 21 days[31]; (4) they had mixed first and second vaccine brands where the second dose was given on or before 7th May 2021[32–34]; (5) the interval between their second and booster vaccinations was less than 90 days[35]; (6) they had conflicting vaccination records or a situation code attached to any vaccination indicating that the vaccination was not given[36]. We applied general quality checks, including removing people from the analysis who had nonsensical dates of birth or death (for details see https://github.com/BHFDSC/CCU002_06).

All eligible people were considered for the first dose analysis. Those who received a first dose of ChAdOx1, BNT-162b2, or mRNA1 were included in the second dose analyses. People who received the same vaccine brand for their first and second doses were included in the booster vaccination analyses.

Data for the study were extracted on 12th May 2022. The study spanned records from 8th December 2020, the start of the UK's vaccine rollout, to 23rd January 2022, the latest available date to ensure completeness of records from across all datasets at data extraction.

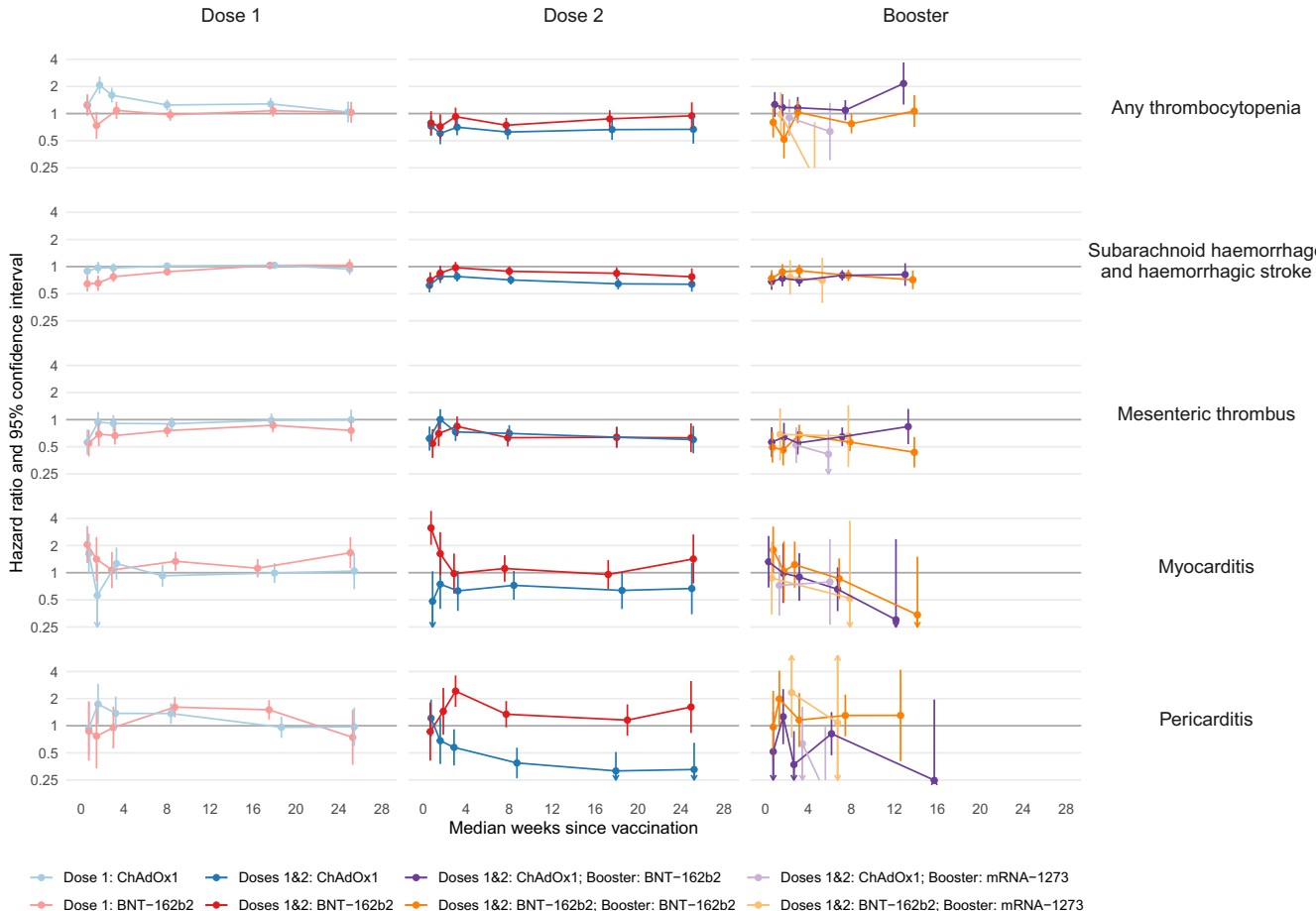

**Fig. 3 | Adjusted hazard ratios (aHRs) and 95% confidence intervals (95% CIs) for other cardiovascular events following COVID-19 vaccination, by dose and brand.** Vertical lines depict 95% CIs; these are not visible when they are very narrow. The number of people eligible for first, second, and booster dose analyses were 45,673,965; 37,249,850 and 35,853,120, respectively. The number of people who received a first dose of ChAdOx1, BNT-162b2 and mRNA-1273 were 19,317,985, 16,846,995, and 1,084,865, respectively; a second dose of ChAdOx1, BNT-162b2 and mRNA-1273 were 18,920,225, 15,961,330, and 971,565, respectively; a booster dose of BNT-162b2 and mRNA-1273 following a primary course of ChAdOx1 were 11,964,635 and 4,153,760 respectively; a booster dose of BNT-162b2 and mRNA-1273 following a primary course of BNT-162b2 were 9,821,835 and 1,914,925, respectively. The numerical values of hazard ratios and 95% CIs are displayed in Supplementary Tables 8, 9, 11, 12, 14, 15, 17, and 18.

Follow-up for first dose vaccination analyses began on 8th December 2020 for all people. For the second and booster vaccination analyses, follow-up commenced on the date of the preceding-dose vaccination. The term "index date" will henceforth denote the start of follow-up for each individual in each analysis.

## Exposures
For the first and second dose analyses, the brands ChAdOx1, BNT-162b2 and mRNA-1273 were analysed separately. For the booster dose analyses, we analysed all combinations of primary courses (ChAdOx1, BNT-162b2, BNT-162b2/mRNA-1273) with booster vaccines (BNT-162b2, mRNA-1273, and BNT-162b2/mRNA-1273). (Supplementary Table 3).

## Confounders
We considered the following confounders: age, sex, ethnic group, Index of Multiple Deprivation (2010 IMD deciles grouped into Deciles 1–4, 5–6, 7–10/missing), smoking status (never/ever, with missing classified as "never"), medical history (including acute myocardial infarction (AMI), diabetes, depression, obesity, cancer, chronic obstructive pulmonary disease (COPD), liver disease, chronic kidney disease, dementia, all stroke, all venous thromboembolic events and thrombophilia), major surgery in the last year, number of unique medical conditions in the last year, prior COVID-19 at index

date, medications taken in the last 90 days (including antiplatelets, blood pressure lowering, lipid-lowering, oral anticoagulants, combined oral contraceptives (COCP) and hormone replacement therapy (HRT)) and clinical vulnerability (clinically extremely vulnerable/clinically vulnerable/neither). The medical history covariates were defined as a diagnosis of the condition before the index date except for diabetes which was additionally defined as a record of diabetic medication in the GDPPR data in the 90 days before the index date. Clinical vulnerability was defined on 8th December 2020. People were flagged as "clinically extremely vulnerable" using the SNOMED code 1300561000000107[37], and "clinically vulnerable" by identifying component conditions as applied in Table 3 of the COVID-19 chapter of the Green Book[30]. Except for sex, clinical vulnerability and ethnic group, all other covariates were updated at individual-specific index dates for the dose 1, dose 2 and booster analyses. History of confirmed COVID-19 diagnosis was ascertained using established algorithms that combine information from SGSS, HES-APC, SUS, ONS deaths registry[38].

## Outcomes
Eleven cardiovascular outcomes were analysed: AMI, ischaemic stroke, lower limb deep venous thrombosis (DVT), pulmonary embolism (PE), intracranial venous thrombosis (ICVT), mesenteric thrombus, portal vein thrombosis (PVT), any thrombocytopenia, subarachnoid

haemorrhage & haemorrhagic stroke (SAH & HS), myocarditis and pericarditis. In addition, two composite outcomes were analysed: composite arterial (AMI, ischaemic stroke and other arterial embolism) and composite venous (PE, DVT, ICVT and PVT). We selected the earliest date of outcome event on or after index date from GDPPR, SUS, HES-APC and ONS deaths registry. We considered only the first/primary position from HES-APC and SUS and used the underlying cause from the death data, to differentiate acute, new events, from prevalent conditions. Further, we had previously found that aHRs for outcomes recorded as primary or secondary reason for admission or death were consistent with those from analyses of outcomes in the primary position[5].

## Statistical analyses

Eligibility criteria and index dates for each vaccination course are detailed in Supplementary Table 3. We censored follow-up at the earliest of death, outcome event of interest, receipt of another vaccine brand, 26 weeks since the vaccination under consideration[9], or the study end date. Baseline demographic and clinical characteristics were detailed for each study cohort, and the number of outcome events and person-years of follow-up were quantified both before and after each vaccination, with incidence rates expressed per 100,000 person-years. Following the NHSE statistical disclosure control process, any counts less than 10 are presented as "10" and all numbers above 10 are rounded to the nearest 5. Percentages and incidence rates were calculated using rounded counts. Therefore, incidence rates corresponding to event counts less than 10 should be regarded as upper bounds and interpreted cautiously.

We analysed the time since vaccination to the first event for each outcome by fitting Cox models with a calendar time scale, with 8th December 2020 as time zero for the first dose analysis and the date of the previous dose as time zero for the second and booster dose analyses. We estimated aHRs and corresponding 95% CIs comparing follow-up after first, second and booster vaccine doses with follow-up before or without the corresponding vaccine dose for time intervals since vaccination: 1 weeks, 2 weeks, 3–4 weeks, 5–12 weeks, 13–24 weeks, and 25–26 weeks after vaccination. We expect that on average, and in the absence of bias, 5% of 95% CIs will exclude the true value of the aHR. Each comparison group included only individuals eligible to receive the vaccine brand and dose under consideration. For example, booster doses can only be received by individuals who have received a primary vaccine course. In the absence of any events in one of these intervals, we consolidated the periods into 1–4 weeks and 5–26 weeks after vaccination. For categorical confounders where fewer than two people in any category experienced an event, we combined categories where feasible and removed the confounder if not feasible. Furthermore, we stratified all models by region to account for between-region variation.

For computational feasibility, our analysis datasets comprised people who experienced the outcome ('cases') during follow-up and a randomly selected subsample with size twenty times the number of cases of those who did not experience the outcome. We applied inverse probability weights to account for this sampling method and used robust standard errors to compute confidence intervals. For each outcome and vaccination course, we estimated hazard ratios adjusting for (i) age and sex; and (ii) all measured confounders (maximally-adjusted).

We performed subgroup analyses for composite arterial and venous outcomes, by age group, ethnic group, prior history of the event (arterial: AMI, ischaemic stroke and other arterial embolism; venous: PE, DVT, ICVT, PVT and other DVT), prior history of confirmed COVID-19 diagnosis, and sex. We repeated the maximally-adjusted analyses for myocarditis and pericarditis, censoring follow-up on 17th May 2021, the date of the CDC's public announcement of their potential associations with COVID-19 mRNA vaccines.

## Software

Analyses used SQL and Python (in Databricks, version 3.68), and RStudio (Professional) Version 1.3.1093.1 driven by R Version 4.0.3 (10th October 2020).

## Ethical approval and information governance

The North East – Newcastle and North Tyneside 2 research ethics committee provided ethical approval for the CVD-COVID-UK/COVID-IMPACT research programme (REC No 20/NE/0161) to access, within secure trusted research environments, unconsented, whole-population, de-identified data from electronic health records collected as part of patients' routine healthcare.

## Reporting summary

Further information on research design is available in the Nature Portfolio Reporting Summary linked to this article.

## Data availability

Data used in this study are available in NHS England's Secure Data Environment service for England for England, but as restrictions apply are not publicly available (https://digital.nhs.uk/services/secure-data-environment-service). The CVD-COVID-UK/COVID-IMPACT programme led by the BHF Data Science Centre (https://bhfdatasciencecentre.org/) received approval to access data in the NHS England's SDE service for England from the Independent Group Advising on the Release of Data (IGARD) (https://digital.nhs.uk/about-nhs-digital/corporate-information-and-documents/independent-group-advising-on-the-release-of-data) via an application made in the Data Access Request Service (DARS) Online system (reference: DARS-NIC-381078-Y9C5K; https://digital.nhs.uk/services/data-access-request-service-dars/dars-products-and-services). The CVD-COVID-UK/COVID-IMPACT Approvals & Oversight Board (https://bhfdatasciencecentre.org/areas/cvd-covid-uk-covid-impact/) subsequently granted approval to this project to access the data within NHS England's SDE service for England. The de-identified data used in this study were made available to accredited researchers only. Those wishing to access the data should contact bhfdsc@hdruk.ac.uk.

## Code availability

All code and code lists are shared openly for review and re-use under an MIT open license (https://github.com/BHFDSC/CCU002_06/).

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

## Acknowledgements

This work was supported by the Longitudinal Health and Wellbeing COVID-19 National Core Study (UKRI Medical Research Council MC_PC_20030 and MC_PC_20059); and by the CONVALESCENCE long COVID study, funded by the UK National Institute for Health and Care Research (COVID-LT-009). This study was also supported by core funding from the: British Heart Foundation (RG/18/13/33946), NIHR Cambridge Biomedical Research Centre (BRC-1215-20014; NIHR203312) [*], Cambridge BHF Centre of Research Excellence (RE/18/1/34212), BHF Chair Award (CH/12/2/29428) and by Health Data Research UK, which receives its funding from HDR UK Ltd (HDR-9006), which is funded by the UK Medical Research Council, Engineering and Physical Sciences Research Council, Economic and Social Research Council, Department of Health and Social Care (England), Chief Scientist Office of the Scottish Government Health and Social Care Directorates, Health and Social Care Research and Development Division (Welsh Government), Public Health Agency (Northern Ireland), British Heart Foundation and the Wellcome Trust. The British Heart Foundation (BHF) Data Science Centre, led by Health Data Research (HDR) UK (BHF Grant no. SP/19/3/34678, awarded to HDR UK) also supported this work. This study made use of de-identified data held in NHS England's Secure Data Environment service for England and made available via the BHF Data Science Centre's CVD-COVID-UK/COVID-IMPACT consortium. This work used data provided by patients and collected by the NHS as part of their care and support. We would also like to acknowledge all data providers who make health relevant data available for research. The BHF Data Science Centre funded co-development (with NHS England) of the Secure Data Environment service for England, provision of linked datasets, data access, user software licenses, computational usage, and data management and wrangling support, with additional contributions from the HDR UK Data and Connectivity component of the UK Government Chief Scientific Adviser's National Core Studies programme to coordinate

national COVID-19 priority research. Consortium partner organisations funded the time of contributing data analysts, biostatisticians, epidemiologists, and clinicians. Further support came from the Con-COV team funded by the Medical Research Council (grant number: MR/V028367/1) and the ADR Wales programme, part of the ADR UK investment, which unites expertise from Swansea University Medical School, WISERD at Cardiff University, and Welsh Government analysts. ADR UK is funded by the Economic and Social Research Council (ESRC), part of UK Research and Innovation. This research was also supported by ESRC funding, including Administrative Data Research Wales (ES/W012227/1). S.I. was funded by the International Alliance for Cancer Early Detection, a partnership between Cancer Research UK C18081/A31373, Canary Center at Stanford University, the University of Cambridge, OHSU Knight Cancer Institute, University College London and the University of Manchester. S.I. and Y.L. are supported by Cancer Research UK EDDPMA-May22\100062. A.B. has received funding from NIHR (COV-LT2-0043) as PI of the STIMULATE-ICP study. V.W. is supported by the Medical Research Council Integrative Epidemiology Unit at the University of Bristol [MC_UU_00032/03]. R.D. and J.A.C.S. are supported by the NIHR Bristol Biomedical Research Centre (NIHR203315) and by Health Data Research UK South-West (HDRUK2023.0022). A.M.W. and JACS are supported by the National Institute for Health Research (NIHR) (NIHR135073). A.M.W. is supported by the BHF Data Science Centre (HDRUK2023.0239) and as an NIHR Research Professor (NIHR303137). A.M.W. conducted this research whilst part of the BigData@Heart Consortium, funded by the Innovative Medicines Initiative-2 Joint Undertaking under grant agreement No 116074 and whilst supported by the BHF-Turing Cardiovascular Data Science Award (BCDSA\100005). The views expressed are those of the author(s) and not necessarily those of NIHR or the Department of Health and Social Care.

## Author contributions

Author contributions are reported below in line with the Contributor Roles Taxonomy (CRediT). Conceptualisation: J.A.C.S., A.M.W., V.W., W.N.W., S.I., T.-L.N. Methodology: J.A.C.S., A.M.W., V.W., W.N.W., S.I., T.-L.N., A.A., E.H. Software: S.I., T.-L.N., E.H., S.K. Validation: S.I., T.-L.N., F.T. Formal analysis: S.I., T.-L.N. Investigation: S.I., T.-L.N. Resources: C.S. Data curation: S.I., T.-L.N. Writing - Original Draft: S.I., T.-L.N., Y.L., W.N.W., J.A.C.S., A.M.W., V.W. Writing - Review & Editing: S.I., T.-L.N., F.T., Y.L., H.A., A.A., E.H., R.D., S.K., S.D., A.B., K.K., C.S., W.N.W., J.A.C.S., A.M.W., V.W. Visualisation: S.I., T.-L.N., Y.L., W.N.W., J.A.C.S., A.M.W., V.W. Project administration: S.I., T.-L.N., J.A.C.S., A.M.W., V.W. Funding acquisition: J.A.C.S., A.M.W., C.S.

## Competing interests

K.K. was chair of the ethnicity subgroup of the UK Scientific Advisory Group for Emergencies (SAGE) and was a member of SAGE. C.S. is Director of the BHF Data Science Centre (whose main funding support comes from the British Heart Foundation) and Chief Scientist and Deputy Director at Health Data Research UK, the UK's Institute for Health Data Science. She played a key role in co-developing NHS England's national secure data environment and leads the CVD-COVID UK/COVID-IMPACT Consortium, which enabled this work. W.W. is supported by the Chief Scientists Office (CAF/01/17) and Stroke Association (SA CV 20100018). W.W. has given expert testimony to UK courts. The remaining authors declare no competing interests.

## Additional information

[1]British Heart Foundation Cardiovascular Epidemiology Unit, Department of Public Health and Primary Care, University of Cambridge, Cambridge, UK. [2]Centre for Cancer Genetic Epidemiology, University of Cambridge, Cambridge, UK. [3]Victor Phillip Dahdaleh Heart and Lung Research Institute, University of Cambridge, Cambridge, UK. [4]Department of Population Health Sciences, Bristol Medical School, University of Bristol, Bristol, UK. [5]Population Data Science, Swansea University Medical School, Faculty of Medicine, Health, and Life Science, Swansea University, Swansea, Wales, UK. [6]NIHR Bristol Biomedical Research Centre, Bristol, UK. [7]Health Data Research UK South-West, Bristol, UK. [8]Health Data Research UK, London, UK. [9]Institute of Health Informatics, University College London, London, UK. [10]University College London Hospitals Biomedical Research Centre, University College London, London, UK. [11]BHF Accelerator, London, UK. [12]British Heart Foundation Data Science Centre, Health Data Research UK, London, UK. [13]Diabetes Research Centre, University of Leicester, Leicester, UK. [14]Centre for Clinical Brain Sciences, University of Edinburgh, Edinburgh, UK. [15]National Institute for Health and Care Research Blood and Transplant Research Unit in Donor Health and Behaviour, University of Cambridge, Cambridge, UK. [16]British Heart Foundation Centre of Research Excellence, University of Cambridge, Cambridge, UK. [17]Health Data Research UK Cambridge, Wellcome Genome Campus and University of Cambridge, Cambridge, UK. [18]Cambridge Centre for AI in Medicine, Cambridge, UK. [19]MRC Integrative Epidemiology Unit, Bristol, UK. [20]Department of Surgery, University of Pennsylvania Perelman School of Medicine, Philadelphia, Pennsylvania, USA. [21]These authors contributed equally: Samantha Ip, Teri-Louise North. [22]These authors jointly supervised this work: Jonathan A. C. Sterne, Angela M. Wood, Venexia Walker. ✉e-mail: hyi20@cam.ac.uk

## the CVD-COVID-UK/COVID-IMPACT Consortium

Samantha Ip [1,2,3,21] ✉, Teri-Louise North[4,21], Fatemeh Torabi [5], Yangfan Li[1,2,3], Hoda Abbasizanjani [5], Ashley Akbari [5], Elsie Horne[4], Rachel Denholm [4,6,7], Spencer Keene[1,3], Spiros Denaxas[8,9,10,11,12], Amitava Banerjee [9], Kamlesh Khunti[13], Cathie Sudlow [12], William N. Whiteley [12,14], Jonathan A. C. Sterne [4,6,7,22], Angela M. Wood [1,3,12,15,16,17,18,22] & Venexia Walker [4,19,20,22]

## the Longitudinal Health and Wellbeing COVID-19 National Core Study

Samantha Ip [1,2,3,21] ✉, Teri-Louise North[4,21], Elsie Horne[4], Rachel Denholm [4,6,7], Spiros Denaxas[8,9,10,11,12], Jonathan A. C. Sterne [4,6,7,22], Angela M. Wood [1,3,12,15,16,17,18,22] & Venexia Walker [4,19,20,22]

