## [Peer Review File · Nature Communications]

Cohort study of cardiovascular safety of different COVID-19 vaccination doses among 46 million adults in EnglandREVIEWER COMMENTS

Reviewer #1 (Remarks to the Author):

In this paper the authors assess cardiovascular events and COVID-19 vaccination in a large cohort in England. The study covers the period to January 2022 which incorporates the first 3 vaccine doses given. This allows follow-up of up to ~ 24 months after first and second doses and a shorter post 3rd dose follow-up. The study is useful given its large size and the interest in cardiac events post COVID-19 vaccination. I have some specific comments to be addressed.

1. Page 9: Methods – note PHE is now UKHSA.
2. Page 10: please clarify the date the data were extracted as well as the date data went to and comment on whether this is long enough for data to be complete (i.e. Hospital data / death data to be complete as these have lags).
3. Page 10: In the section on confounders it was not clear initially what the index date is. Please make it clear before the start of this section what the index dates are for the first, second and booster dose analyses.
4. Page 10. When looking at second / booster doses comparisons are made within those who have received one dose / two doses rather than also including the unvaccinated. What is the rationale for this? This should be made clear. For example if there is an ongoing long term increased risk after at least one dose then this is only shown for the analysis after the first dose since the second dose would be shown not to add any risk on top of the first dose risk. If, on the other hand, risks after a first dose were transient (<3 weeks) then the effects seen after the second dose would be the same as if compared to unvaccinated.
5. Table S3: I got confused on this paper about the comparator group. Doesn't the comparator for dose 2 include those who don't get ever get a second dose? Similar for the booster analysis.
6. Page 11: The study focuses on estimation rather than hypothesis test (no p-values given). Please do state in the analysis that aHRs are estimated with 95%CI. Please also state that this study is about estimation. The issue I have is that there are a lot of estimates. As such some will, by chance, exclude a value of 1 in the 95% CI. It would be helpful if the authors could indicate what size of effect and corresponding lower end of the 95% CI they would regard as some evidence of a higher rate and what they regard as clear evidence. In the results for example on page 6 1.65 (1.07-2.57) is seen as "somewhat higher" but in the next paragraph a result of 1.74 (1.04-2.91) is not regarded as borderline? I would argue any result with bottom end of a 95% CI below about 1.2 is borderline.
7. Figure1: please clarify that vertical lines are 95% CIs and that when not shown this is because estimates have very tight 95% CIs (I think that is the case). Also did you have a criteria for not showing an estimate? To me the longer interval for d1&2 BNT with a booster mRNA has numbers that are too small given the very wide 95% CI. Similarly in later figures many estimates have huge and uninformative 95% CIs for longer periods after boosters
8. Page 7: The lack of effects for myocarditis and pericarditis when the data were restricted to prior to May 17th 2021 may well be down to low power. This should be noted.
9. The discussion covers well the limitations and explanations of where HRs are low. The issue of the comparability with those who have delayed doses could be covered better. Given that the cox regression has a calendar time scale, the HRs at the longer intervals post second dose (and similarly for boosters) will be being compared to those who for some reason never got a second dose. Not getting a second dose may well be due to factors that make the events more likely (e.g. being in hospital). A similar argument also applies when considering the long intervals after dose 1 (vs unvaccinated).

Reviewer #2 (Remarks to the Author):

The authors presented the cardiovascular events after different COVID-19 vaccination during the first 2 years of the UK vaccination program. The Longitudinal health records consisted of 45.7 million adults in England which is no doubt, with one of the largest studies of COVID-19 vaccine safety in UK. I have some comments and suggestions for the editors and authors to consider.

I think this study is well-conducted and well-written. The authors reported similar findings as previously published (10.1136/bmj.n1931 ; <https://doi.org/10.1038/s41591-021-01630-0>). There is less novelty in terms of the data given that only first 2 years (up to early 2022) of covid 19 vaccination program were captured whilst the most pressing question now is the effectiveness and long-term safety of bivalent and subsequent booster doses. However, I agree with the authors that the novelty of this study lies in the representativeness of the whole population data.

I am curious about the choice of methodology. Given the massive data source used in this study, the authors can consider using self-controlled case series (SCCS) study (doi: 10.1002/sim.9325). The uptake rate of COVID-19 vaccines were fast when the program first rolled out, therefore, it would be difficult to quantify the risk of adverse events when compared to non-vaccinated group. The authors may consider adding a post-hoc analysis of SCCS as an internal validation of the current findings.

Dear Reviewers,

Thank you for your comments on our manuscript “Cohort study of cardiovascular safety of different COVID-19 vaccination doses among 46 million English adults” (NCOMMS-24-10984A), which have helped us improve the quality and clarity of our paper.

We have carefully addressed each comment, provided detailed responses below, and made corresponding revisions to the manuscript. We hope that our revisions and responses adequately address the concerns raised.

Reviewer #1 (Remarks to the Author):

In this paper the authors assess cardiovascular events and COVID-19 vaccination in a large cohort in England. The study covers the period to January 2022 which incorporates the first 3 vaccine doses given. This allows follow-up of up to ~ 24 months after first and second doses and a shorter post 3rd dose follow-up. The study is useful given its large size and the interest in cardiac events post COVID-19 vaccination. I have some specific comments to be addressed.

1. Page 9: Methods – note PHE is now UKHSA.

Thank you for pointing out the need to update the acronym from PHE to UKHSA, we have updated the manuscript accordingly.

2. Page 10: please clarify the date the data were extracted as well as the date data went to and comment on whether this is long enough for data to be complete (i.e. Hospital data / death data to be complete as these have lags).

We have added clarification in the “Methods: Study population” section as follows (revised text in bold italics): “***Data for the study were extracted on 12th May 2022.*** The study spanned records from 8th December 2020, the start of the UK’s vaccine rollout, to 23rd January 2022, the latest available date ***to ensure completeness of records from across all datasets at data extraction.***”

3. Page 10: In the section on confounders it was not clear initially what the index date is. Please make it clear before the start of this section what the index dates are for the first, second and booster dose analyses.

We have added the definition of index date in the “Methods: Study population” section as follows (revised text in bold italics): “Follow-up for first dose vaccination analyses began on 8th December 2020 for all people. For the second and booster vaccination analyses, follow-up commenced on the date of the preceding-dose vaccination. ***The term “index date” will henceforth denote the start of follow-up for each individual in each analysis.***”

4. Page 10. When looking at second / booster doses comparisons are made within those who have received one dose / two doses rather than also including the unvaccinated. What is the rationale for this? This should be made clear. For example if there is an ongoing long term increased risk after at least one dose then this is only shown for the analysis after the first dose since the second dose would be shown not to add any risk on top of the first dose risk. If, on the other hand, risks after a first dose were transient (<3 weeks) then the effects seen after the second dose would be the same as if compared to unvaccinated

We have now expanded paragraph 2 in the “Methods: Statistical Analyses” section to explain that the comparator group was selected to include those only eligible for the second / booster doses, as follows (revised text in bold italics): “***We expect that on average, and in the absence of bias, 5% of***

95% CIs will exclude the true value of the aHR. Each comparison group included only individuals eligible to receive the specific vaccine brand and dose under consideration. For example, booster doses can only be received by individuals who have received a primary vaccine course.

5. Table S3: I got confused on this paper about the comparator group. Doesn't the comparator for dose 2 include those who don't get ever get a second dose? Similar for the booster analysis. We have clarified this by expanding the caption of Supplementary Table 3, using similar wording to that in response to comment 4 above (revised text in bold italics): "Supplementary Table 3: Definitions of the period after-vaccination and the corresponding period before or without vaccination in eligible individuals. **Each comparison group included only individuals eligible to receive the vaccine brand and dose under consideration.**"

We revised the column headers in Table 2 and Supplementary Tables 4-7 to explicitly specify the vaccine dose received by each group under comparison for enhanced clarity.

6. Page 11: The study focuses on estimation rather than hypothesis test (no p-values given). Please do state in the analysis that aHRs are estimated with 95%CI. Please also state that this study is about estimation. The issue I have is that there are a lot of estimates. As such some will, by chance, exclude a value of 1 in the 95% CI. It would be helpful if the authors could indicate what size of effect and corresponding lower end of the 95% CI they would regard as some evidence of a higher rate and what they regard as clear evidence. In the results for example on page 6 1.65 (1.07-2.57) is seen as "somewhat higher" but in the next paragraph a result of 1.74 (1.04-2.91) is not regarded as borderline? I would argue any result with bottom end of a 95% CI below about 1.2 is borderline. Thank you - we have clarified this issue as follows:

In the introduction (revised text in bold italics): "We used Cox regression to estimate adjusted hazard ratios (aHRs) **and corresponding 95% confidence intervals (95% CIs)** in time intervals since vaccination, adjusted for a wide range of co-morbidities, age, sex, and prior COVID-19".

In the section 'Results: COVID-19 vaccination and arterial, venous, and other thrombotic events' (revised text in bold italics): "We used Cox models to estimate adjusted hazard ratios (aHRs) **and corresponding 95% CIs,** comparing the incidence of thrombotic and cardiovascular events after first, second and booster vaccine doses with the incidence before or without the corresponding vaccine dose, adjusting for a wide range of potential confounding factors".

In the section 'Statistical methods' (revised text in bold italics): "We estimated aHRs **and corresponding 95% CIs** comparing follow-up after first, second and booster vaccine doses with follow-up before or without the corresponding vaccine dose for time intervals since vaccination: 1 weeks, 2 weeks, 3-4 weeks, 5-12 weeks, 13-24 weeks, and 25-26 weeks after vaccination. We expect that, on average and in the absence of bias, 5% of 95% CIs will exclude the true value of the aHR."

We are not able to define the size of effect or lower limit of the 95% CI we consider indicative of higher rate. As the reviewer is aware, many of the 95% CIs that we report clearly exclude the null HR of 1 and corresponding p values are small. For common outcomes, the main issue in interpretation is the potential for bias (for example, due to unmeasured confounding) rather than sampling variation. For rare outcomes such as intracranial venous thrombosis or thrombocytopenia, confidence intervals are wider and interpretation of findings should relate both to the extent of sampling variation, the consistency of our findings with other evidence about rare harms of COVID-19 vaccination, and the specificity of our findings to these known harms. We hope that these issues are adequately addressed in the limitations paragraph in the discussion section of our manuscript.

7. Figure 1: please clarify that vertical lines are 95% CIs and that when not shown this is because estimates have very tight 95% CIs (I think that is the case). Also did you have a criteria for not showing an estimate? To me the longer interval for d1&2 BNT with a booster mRNA has numbers that are too small given the very wide 95% CI. Similarly in later figures many estimates have huge and uninformative 95% CIs for longer periods after boosters

We have revised the caption for Figure 1 (and similarly for Figures 2 and 3): “Figure 1: ***Adjusted hazard ratios (aHRs) and 95% confidence intervals (95% CIs)*** for arterial thromboses following COVID-19 vaccination, ***by dose and brand. Vertical lines depict 95% CIs; these are not visible when they are very narrow.***”

We have also added a footnote to Figure 1: “***There were no AMI events during weeks 13-26 after mRNA-1273 booster vaccination, so follow-up is grouped as 1-4 and 5-26 weeks post-vaccination.***”

We aimed to display all estimated aHRs, as comprehensively as possible, and did not have a criterion for not showing an estimate. We are comfortable that the wide 95% CIs for d1&2 BNT with a booster mRNA clearly communicate that the corresponding estimated aHRs (for example 2.29 (95% CI 0.67, 7.82) for ischaemic stroke) are consistent with a wide range of values including 1. Please also note that the scale of the y-axes in Figures 1-3 are only from 0.25 to 4.

8. Page 7: The lack of effects for myocarditis and pericarditis when the data were restricted to prior to May 17th 2021 may well be down to low power. This should be noted.

We have now expanded on this point in the Results and Discussion sections. In the Results (page 7, paragraph 1) we now write (revised text in bold italics): “***Note that the smaller number of events in this sensitivity analyses meant that aHRs were less precisely estimated than in the main analyses.***” In the paragraph on limitations of our study in the discussion section we now write: “Third, ascertainment of some outcomes may have been influenced by public announcements from regulatory agencies, such as the European Medicines Agency Pharmacovigilance Risk Assessment Committee announcement (11) or the CDC announcement on myocarditis (12,13). This was addressed in sensitivity analyses for myocarditis and pericarditis, censoring follow-up at the time of public announcements of these adverse effects of vaccination, ***although the shorter follow-up times and corresponding smaller numbers of events in the restricted analyses meant that aHRs were estimated with reduced precision.***”

9. The discussion covers well the limitations and explanations of where HRs are low. The issue of the comparability with those who have delayed doses could be covered better. Given that the cox regression has a calendar time scale, the HRs at the longer intervals post second dose (and similarly for boosters) will be being compared to those who for some reason never got a second dose. Not getting a second dose may well be due to factors that make the events more likely (e.g. being in hospital). A similar argument also applies when considering the long intervals after dose 1 (vs unvaccinated).

We have further clarified these limitations in our points made about residual confounding, time-varying confounders and censoring our analyses at 26 weeks as follows (revised text in bold italics): “This study has several limitations. First, residual confounding, ***including that linked to delayed vaccination in high-risk individuals,*** may persist despite extensive adjustments for available covariates. We were able to identify some, but not all people who were clinically vulnerable (and hence might have been eligible for earlier vaccination): for example, younger adults in long-stay settings could not be reliably identified. Second, we did not adjust for potential confounding by time-varying post-baseline factors that may have influenced receipt of vaccination ***and the outcomes of interest:*** for example, development of respiratory symptoms ***or being admitted into hospital*** leading to postponement of vaccination. Such confounding may explain estimated lower

hazard ratios soon after vaccination (10).

[...]

Fifth, we **restricted** follow-up to 26 weeks after vaccination to prevent an influence of subsequent vaccinations on estimated associations **and limit the impact of delayed vaccination on our findings.** Horne et al. demonstrated selection bias in estimated HRs for non-COVID-19 death arising from deferred next-dose vaccination in people with a recent confirmed COVID-19 diagnosis or in poor health (9).”

Reviewer #2 (Remarks to the Author):

The authors presented the cardiovascular events after different COVID-19 vaccination during the first 2 years of the UK vaccination program. The Longitudinal health records consisted of 45.7 million adults in England which is no doubt, with one of the largest studies of COVID-19 vaccine safety in UK. I have some comments and suggestions for the editors and authors to consider.

I think this study is well-conducted and well-written. The authors reported similar findings as previously published (10.1136/bmj.n1931 ; <https://doi.org/10.1038/s41591-021-01630-0>). There is less novelty in terms of the data given that only first 2 years (up to early 2022) of covid 19 vaccination program were captured whilst the most pressing question now is the effectiveness and long-term safety of bivalent and subsequent booster doses.

Thank you for your positive view of our study. As noted in our response to Reviewer 1 point 9 above, our analyses were limited to 26 weeks follow-up post-vaccination, to prevent an influence of subsequent vaccinations on estimated associations and limit the impact of delayed vaccination on our findings. We agree that we did not address long-term safety or subsequent booster doses: these issues are beyond the scope of our study. We have noted this in the paragraph on limitations in the discussion section: **“Sixth, we did not address long-term safety of vaccination, or the impact of subsequent booster doses”**. We have also removed **“Long-term”** from our title to more accurately reflect the study's focus.

However, I agree with the authors that the novelty of this study lies in the representativeness of the whole population data.

I am curious about the choice of methodology. Given the massive data source used in this study, the authors can consider using self-controlled case series (SCCS) study (doi: 10.1002/sim.9325). The uptake rate of COVID-19 vaccines were fast when the program first rolled out, therefore, it would be difficult to quantify the risk of adverse events when compared to non-vaccinated group. The authors may consider adding a post-hoc analysis of SCCS as an internal validation of the current findings. The Self-Controlled Case Series (SCCS) method is indeed a viable alternative for analysing our extensive dataset, offering computational efficiency and addressing some of the limitations of traditional cohort designs. We are engaged in ongoing work comparing estimates from the SCCS and cohort designs, which make different assumptions, and have different strengths and weaknesses. In particular, the SCCS method assumes event-independent observation periods, requires medically guided definitions of risk periods and is most suitable for investigating short-term, acute outcomes, due to potential constraints associated with the availability of suitable reference periods. Replicating all our analyses using the SCCS framework and conducting all the necessary sensitivity assessments would go well beyond the scope of our current study.

Our decision to employ a cohort study design was influenced by its broad acceptance and the straightforward nature of its interpretation by a general audience. Cohort designs also enable

estimation of incidence rates (see Table 2) and hence facilitate an understanding of the population-level impact of vaccination on common and rare cardiovascular events.

We added the following to the Discussion section (revised text in bold italics): **“The cohort design of our study allows estimation of incidence rates (see Table 2) and hence facilitates an understanding of the population-level impact of vaccination on common and rare cardiovascular events. However, our findings may be biased by unmeasured confounding, as discussed above. The self-controlled case series (SCCS) method aims to avoid the need to control for time-invariant confounders, but entails assumptions such as event-independent observation periods and medically-informed risk period definitions, and is particularly suited to short-term outcomes. Our results thus complement, and may be triangulated with, previous publications that examined cardiovascular implications of COVID-19 vaccination using the SCCS method (27-29).”**

We hope that our responses and the revisions made to the manuscript address your concerns satisfactorily. We believe these changes have strengthened our paper and we sincerely thank you for your constructive feedback.

Yours sincerely,

Dr. Samantha Ip

REVIEWERS' COMMENTS

Reviewer #1 (Remarks to the Author):

The authors have satisfactorily addressed my comments. I have no further comments.